# Antidyslipidemia Pharmacotherapy in Chronic Kidney Disease: A Systematic Review and Bayesian Network Meta-Analysis

**DOI:** 10.3390/pharmaceutics15010006

**Published:** 2022-12-20

**Authors:** Guangzhi Liao, Xiangpeng Wang, Yiming Li, Xuefeng Chen, Ke Huang, Lin Bai, Yuyang Ye, Yong Peng

**Affiliations:** 1Department of Cardiology, West China Hospital, Sichuan University, 37 Guoxue Street, Chengdu 610041, China; 2Department of Rheumatology and Immunology, West China Hospital, Sichuan University, Chengdu 610041, China; 3West China School of Medicine, Sichuan University, Chengdu 610041, China

**Keywords:** systematic review, bayesian network meta-analysis, antidyslipidemia pharmacotherapy, chronic kidney disease

## Abstract

Background and Aims: The benefits and safety of antidyslipidemia pharmacotherapy in patients with chronic kidney disease were not well defined so the latest evidence was summarized by this work. Methods: This systematic review and Bayesian network meta-analysis (NMA) included searches of PubMed, Embase, and Cochrane Library from inception to 28 February 2022, for randomized controlled trials of any antilipidaemic medications administered to adults with chronic kidney disease [CKD: defined as estimated glomerular filtration rate (eGFR) ≤ 60 mL/min/1.73 m^2^ not undergoing transplantation], using the Grading of Recommendations Assessment, Development, and Evaluation (GRADE) tool to assess the certainty of the evidence. Results: 55 trials and 30 works of them were included in our systematic review and NMA, respectively. In comparisons with no antidyslipidemia therapy or placebo, proprotein convertase subtilisin/Kexin type 9 inhibitors plus statin (PS) was the most effective drug regimen for reducing all-cause mortality (OR 0.62, 95% CI [0.40, 0.93]; GRADE: moderate), followed by moderate-high intensity statin (HS, OR 0.76, 95% CI [0.60, 0.93]; I^2^ = 66.9%; GRADE: moderate). PS, HS, low-moderate statin (LS), ezetimibe plus statin (ES), and fibrates (F) significantly decreased the composite cardiovascular events. The subgroup analysis revealed the null effect of statins on death (OR 0.92, 95% CI [0.81, 1.04]) and composite cardiovascular events (OR 0.94, 95% CI [0.82, 1.07]) in dialysis patients. Conclusion: In nondialysis CKD patients, statin-based therapies could significantly and safely reduce all-cause death and major composite cardiovascular events despite the presence of arteriosclerotic cardiovascular disease and LDL-c levels. Aggressive medication regimens, PS and HS, appeared to be more effective, especially in patients with established CAD.

## 1. Introduction

With an estimated prevalence of 13.4% globally [1], chronic kidney disease (CKD) is currently a public health problem worldwide that imposes tremendous medical and financial burdens on society and healthcare systems. These patients have an absolute risk for cardiovascular events similar to that of persons with established coronary artery disease (CAD), and the risk increases linearly as the estimated glomerular filtration rate (eGFR) decreases below 60 mL/min/1.73 m^2^ [2]. Given the elevated cardiovascular risk and evidence-based benefits of lipid management in general people, antidyslipidemia pharmacotherapy was assumed to be a reasonable way to improve clinical outcomes of CKD. However, unlike non-CKD populations, they frequently exhibit a characteristic lipid profile of hypertriglyceridemia but mostly normal or low LDL-c [3], and the benefit of hypolipidemia pharmacotherapy becomes smaller as eGFR declines, with no evidence of benefit but potential drug-related adverse effects due to increased blood concentrations of compounds on dialysis [4]. Therefore, controversy surrounds lipid-lowering pharmacotherapy in CKD. In addition, as malnutrition and inflammation are also essential elements to consider, the relationship between serum lipids, especially LDL-c, and arteriosclerotic cardiovascular disease is more challenging to clarify in CKD.

Prior reviews and meta-analyses pooling evidence from RCTs have shown the benefits of statin therapy (with or without ezetimibe) for mortality and cardiovascular outcomes in CKD patients not requiring dialysis [5]. Based on these works, initiation of lipid-lowering medication in CKD was recommended in the latest guidelines [4,6,7]. However, few of the CKD population receive statin therapy in clinical practices, and the effect of other hypolipidemic medications was not well defined. The recommendations of the optimal drug dose and the prescription were also omitted. More RCTs have been published in recent years. Consequently, in light of the abovementioned controversial issues meriting consideration and newly reported evidence, we undertook this systematic review and network meta-analysis (NMA) to summarize the benefits and safety of hypolipidaemic medications and further provide the optimal regimens of the various treatments for CKD patients through network comparisons.

## 2. Methods

### 2.1. Study Design

This systematic review and NMA had been registered on PROSPERO (CRD42022322729). We performed and reported this work following the Cochrane Collaboration guidelines and Preferred Reporting Items for Systematic Reviews and Meta-Analysis Extension (PRISMA) statement for systematic reviews incorporating network meta-analyses for health care interventions [8,9].

### 2.2. Data Sources and Searches

We searched the PubMed, Embase, and Cochrane Library databases from inception to 28 February 2022, with English as the language restriction. Clinical trial registries (www.clinicaltrials.gov, www.clinicaltrialsregister.eu, accessed on 29 February 2022), and references of the relevant articles, especially reviews, were also searched and reviewed to supplement the identified citations. Searches included terms relating to CKD, lipid-lowering medication, and randomized controlled trials (RCTs; Appendix A). Three reviewers (LG, HK, and LX) used EndNote X9 (Clarivate Analytics, Philadelphia, PA, USA) to screen titles and abstracts, after which they read the full-text manuscripts and extracted data on study identifiers, study design and setting, participant characteristics at baseline, intervention, and outcomes. Discrepancies were resolved by discussion.

### 2.3. Study Selection

Eligible studies had to meet the following criteria: (1) RCTs irrespective of year of publication; (2) participants: adults aged ≥ 18 years old with CKD (defined as eGFR < 60 mL/min/1.73 m^2^) regardless of comorbid clinically evident atherosclerosis and hyperlipidemia; (3) interventions: monotherapy or combination of any antidyslipidaemic pharmacotherapy and placebo/diet therapy; (4) outcomes: we deemed all-cause death as the critical outcome of interest; composite cardiovascular events (CV events, the definitions of the outcomes were taken as reported in the individual studies (Appendix A) were the important but not critical outcomes; cardiovascular (CV) death, stroke, myocardial infarction (MI) and a reduction in LDL-c, safety outcomes including the change in eGFR and major drug-related adverse events (any muscle-related abnormalities, aspartate aminotransferase (AST)/alanine aminotransferase (ALT) ≥2 times of upper limit) were deemed as less important outcomes. Subgroup analyses were planned to be performed in patients receiving dialysis therapy and with coronary artery disease. Only studies with at least a 3-month follow-up period were included, as those with a shorter duration would not allow the detection of significant changes in LDL-c. And only those with a follow-up duration ≥ 1 year can be included in the analysis of all-cause death and cardiovascular outcomes. To avoid minor study effects and generate more robust evidence, we selected studies enrolling at least 50 patients. In addition, we excluded trials for the following reasons: trials that used quasirandom methods; and studies enrolling mostly patients who underwent kidney transplants or were on the transplant waiting list. We have made some adjustments to the existing protocols: the medications included in this work finally were not limited to PCSK9i, statins, ezetimibe, and fibrates; we also explore the clinical outcomes of other hypolipidemic drugs by reviewing the RCTs related.

### 2.4. Data Extraction and Quality Assessment

Data extraction was performed independently by two authors (LG and WX) with a prespecified data extraction form incorporating information about studies, baseline characteristics of the participants, and outcomes-related data. We made efforts to contact the authors if the important data were not reported. The discrepancies were resolved by consensus or in consultation with a third reviewer (LY). The risk of bias assessment at the study level was performed by LG and WX independently using Rob2 [10], a revised Cochrane risk-of-bias tool for RCTs, with discrepancies resolved by LY. Then, we used the GRADE approach for entire networks, providing the framework for rating the certainty of the evidence of each paired comparison as high, moderate, low, or very low [11,12]. We also assess confidence in the results utilizing the Confidence-in-Network-Meta-Analysis (CINeMA) approach [13].

### 2.5. Data Synthesis and Analysis

The odds ratio (OR), the mean difference (MD), and their corresponding 95% confidence intervals (95% CI) were calculated for dichotomous outcomes and continuous outcomes with the random-effect model. Traditional pairwise meta-analysis by Review Manager 5.3 and network meta-analysis (NMA) with a Bayesian model through Markov chain Monte Carlo (MCMC) simulation by the R package GeMTC were carried out sequentially. The node-splitting method was applied to evaluate the inconsistency between direct and indirect comparisons. Inconsistency was deemed nonsignificant when the *p*-value was >0.05 for the comparison between direct and indirect effects. We quantified the proportion of variation in the meta-analyses due to clinical and methodological heterogeneity using I^2^ (Higgins 2002), the value of which was classified as might not be important (0% to 40%), may represent substantial heterogeneity (30% to 60%) and considerable heterogeneity (75% to 100%). Convergence was assessed through visual inspection of the Brooks-Gelman-Rubin diagnostic, with convergence assumed to have occurred when the ratio of between- and within-chain variability was stable at approximately one. Varying iterations and burn-in periods were used to ensure convergence, with burn-in periods discarded from the analysis. With Markov chain Monte Carlo (MCMC) modeling, the relative ranking probability of each treatment group was estimated, and then “Ranko grams” with the surface under the cumulative ranking curve (SUCRA) were reported to provide a comparative hierarchy of efficacy of the treatment groups. Prespecified subgroup analyses for patients with CKD and those receiving dialysis were performed. In addition, regression analysis and sensitivity analysis were performed to investigate the influence of serum LDL-c level and the presence of established arteriosclerotic cardiovascular disease on the results. Publication bias was assessed using a funnel diagram with Stata MP software version 15.0. To test the robustness of the results, we also conducted the analyses utilizing the frequentist method.

## 3. Results

### 3.1. Description of Trials

Of the 6123 screened articles, 146 relevant full-text articles were selected after the removal of irrelevant trials, duplicate articles, reviews, protocols, and unavailable abstracts. After screening, 55 trials involving proprotein convertase subtilisin/Kexin type 9 inhibitors (PCSK9i), statins, ezetimibe, fibrates, bile acid sequestrants, niacin, and polyunsaturated fatty acids were finally included in our systematic review (Appendix A). Given the practical clinical applications and the numbers of eligible RCTs, the NMA work included only the first 4 agents. Consequently, 30 works that included 45,627 patients and involved 5 medication regimens—PCSK9i under background use of a moderate-high intensity statin (PS), statins of low-moderate intensity (LS: <40 mg of atorvastatin) and moderate-high intensity (HS: 40–80 mg of atorvastatin or the highest dose patients could tolerant), the combination of ezetimibe and statin (ES) and fibrates (F)—were available for quantitative analyses [14,15,16,17,18,19,20,21,22,23,24,25,26,27,28,29,30,31,32,33,34,35,36,37,38,39,40,41,42,43]. The definition of CKD differed among the studies, with most applying eGFR below 60 mL/min/1.73 m^2^ as the criteria. Although a small number of studies enrolling some patients with a higher eGFR, we still included them as the majority of patients met our requirement. The average follow-up duration was 51.8 months. The inclusion criteria and major findings of all 55 studies are summarized and presented in Appendix A.

### 3.2. Patient Characteristics

The baseline characteristics of patients enrolled in the NMA are shown in Table 1. The cohorts consisted of 58.0% male patients, with an average age of 65.4 years. The frequently reported comorbid conditions were diabetes, hypertension, and arteriosclerotic cardiovascular disease, and other major medication treatments included but were not limited to antiplatelet medicines, antihypertensive agents such as angiotensin-converting enzyme inhibitors/angiotensin receptor blockers (ACEIs/ARBs), beta-blockers and calcium channel blockers (CCBs). Where data were available, the baseline LDL-c, high-density lipoprotein cholesterol (HDL-c) and triglyceride (TG) levels were 3.0, 1.2, and 1.9 mmol/L, respectively. Only 3 trials studied the effects of statins in dialysis patients.

### 3.3. Risk of Bias in Individual Trials and the Grade of Evidence

The risk of bias at the study level is summarized in Appendix A. Eleven trials were at high risk of bias primarily because of performance and detection bias. Except for Stegmayr’s study [37], the trials pooled to conduct the NMA were assessed as works of moderate-high quality. The details of evidence evaluation utilizing GRADE and CINeMA approach are both available in Appendix A.

### 3.4. Outcomes of Interest

#### 3.4.1. All-Cause Death

A total of 20 studies reported information about the effect of lipid-lowering therapies on all-cause mortality in CKD patients, and finally, the data of 18 studies that involved 41,213 individuals were available for NMA [16,18,19,20,21,22,23,29,33,34,35,36,37,38,40,41,42,44]. Network plots of available direct comparisons and the pooled odds ratios (OR) with 95% confidence intervals (95% CI) corresponding to the pairwise meta-analysis are shown in Figure 1A and Figure 2. The direct comparisons suggested that compared to placebo/no drug therapy, LS (LS: OR 0.88, 95% CI [0.80, 0.97]) and HS (OR 0.54, 95% CI [0.36, 0.83]) were associated with death reduction. Consistently, Bayesian network meta-analysis (Figure 3A) confirmed that compared to placebo/ no drug use, HS (OR 0.76, 95% CI [0.60, 0.93]; I^2^ = 66.9%; GRADE: moderate) and LS (OR 0.89, 95% CI [0.76,0.99]; I^2^ = 0%; GRADE: moderate) could both reduce death risk in CKD patients. In addition, PS also had a similar effect (OR 0.62, 95% [0.40, 0.93]; GRADE: moderate) (indirect comparison). ES (OR 0.94, 95% CI [0.74, 1.1]; I^2^ = 90.6%; GRADE: low) and F (OR 0.84, 95% CI [0.55, 1.2]; I^2^ = 0%; GRADE: low) both failed to reduce mortality. Figure 3C and Figure 4A show the league table for all these therapies and surface under the cumulative ranking (SUCRA) plot based on the cumulative probability of each treatment. PS was ranked as the best treatment for having the highest probability of reducing the all-cause mortality risk (SUCRA, 94%), followed by HS (SUCRA, 73%). There was no relevant inconsistency (Appendix A) or significant publication bias (Appendix A).

#### 3.4.2. The Composite Cardiovascular Events

22 studies reported composite cardiovascular events with various definitions (Appendix A), and 20 studies were pooled to perform the NMA (Figure 1B) [14,16,17,18,19,20,21,22,23,29,33,34,35,36,37,38,39,40,41,44]. Compared with placebo or nonmedication management, PS (OR 0.45, 95% CI [0.24, 0.74]; GRADE: moderate) (no direct comparison), HS (OR 0.54, 95% CI [0.39, 0.73]; I^2^ = 0%; GRADE: moderate), LS (OR 0.72, 95% CI [0.57, 0.85]; I^2^ = 66.7%; GRADE: moderate) and F(OR 0.63, 95% CI [0.39,0.98]; I^2^ = 0%; GRADE: moderate) could reduce the risk of composite cardiovascular events, while studies on ES provided low-certainty evidence of this benefit (OR 0.68, 95% CI [0.47, 0.91]; I^2^ = 89.3%]; GRADE: low) (Figure 3B). Figure 3D and Figure 4B show the league table for all these therapies and surface under the cumulative ranking (SUCRA) plot based on the cumulative probability of each treatment. In line with the result of all-cause mortality, PS had the highest probability of reducing the occurrence of composite cardiovascular events (SUCRA, 92%), followed by HS (SUCRA, 74%). There was no relevant inconsistency (Appendix A). The funnel plot suggested the existence of publication bias (Appendix A).

#### 3.4.3. CV Death, Stroke, MI, and LDL-c Reduction

Only 11 (Appendix A) and 13 studies (Appendix A) were available for inclusion in the pooled analyses for CV death and stroke (Figure 1C,D). Not enough data were available for myocardial infarction (MI) analysis. No evidence of moderate-high grade supported the significant association between the therapy and reduction of CV death and stroke event (Appendix A). 44 works studied the LDL-reducing effect of hypolipidemic treatment, while only 14 of them (Appendix A) provided systematic reports on the outcome of LDL-c reduction (Figure 1E). According to the trials without a high risk of bias, the effects on LDL-c reduction of these regimens remained consistent with those in the general population. Fibrates were associated with a significant reduction in TG without an obvious effect on LDL-c (MD 0.28, 95% CI [−0.43, 0.96]; GRADE: very low). ES (MD −1.1, 95% CI [−1.5, −0.64]; GRADE: moderate) and LS (MD −0.87, 95% CI [−1.1, −0.62]; GRADE: moderate) could both exert a strong effect on LDL-c reduction (Appendix A), while E (−0.19, 95% CI [−0.94, 0.55]: GRADE: low) was not associated with a significant reduction in LDL-c in CKD patients. According to the ODYSSEY OUTCOME trial, PS can lead to further LDL-c reduction (MD −1.22, 95% CI [−1.31, −1.13]) than HS.

#### 3.4.4. Estimated Glomerular Filtration Rate

29 trials reported antilipidemic treatment effects on renal function, while we failed to perform a quantitative analysis, as few of them reported the renal outcome systematically. Studies of moderate-high quality revealed no significant association between the decrease in eGFR and the use of lipid-lowering treatment among CKD patients regardless of baseline renal function. on the contrary, they found that statin-based therapy improved the eGFR slightly or retarded its further decrease in the long-term follow-up [18,19,32,41,46]. Compared to LS, an aggressive antilipidemic method using HS augments the reno-protective property [18,19]. The combination of fenofibric acid and rosuvastatin was found to be associated with a reversible negative impact on eGFR [47].

#### 3.4.5. Major Adverse Events

After excluding trials with a high risk of bias, the data in 23 trials with moderate-high quality were available for pairwise meta-analysis. In these trials reporting the major adverse events, including muscle-related adverse outcomes and ALT/AST elevation, no significant difference was observed between PS and HS alone, ES and LS, F and placebo, or LS and placebo/no drug treatment (Appendix A).

#### 3.4.6. The Benefits and Harms of Other Hypolipidemic Agents and Polyunsaturated Fatty Acids Supplementation

Based on the findings from the related moderate-high quality RCTs [48,49,50,51], supplementation with ω-3/ω-6 could effectively improve lipid metabolism disorder by reducing the LDL/HDL-c ratio without an obvious effect of decreasing LDL-c in dialysis patients. According to the studies with a low-moderate risk of bias investigating the benefits of colestilan in ESRD individuals [52,53,54], the LDL-c-reducing effect of colestilan was significant and even not inferior to simvastatin. During the treatment phase, except for gastrointestinal reactions and some other mild-moderate drug-related adverse events, no serious side effects were observed in dialysis patients. In addition to traditional agents, the application of one new antidyslipidemic medicine, Apabetalone, was discovered to decrease major cardiovascular events by 50% in CKD and diabetes patients with a high burden of arteriosclerotic cardiovascular disease [55] (Appendix A).

#### 3.4.7. Subgroup Analysis, Regressive Analysis for Baseline LDL-c, and Sensitivity Analysis

Only 3 trials focused on the effect of statin treatment in advanced CKD and reported corresponding information systematically. The subgroup analysis in this cohort suggested the null effect of statins on mortality (OR 0.92, 95% [0.81, 1.04]; I^2^ = 0%; GRADE: moderate) and composite cardiovascular event reduction in this group (OR 0.94, 95% CI (0.82, 1.07); GRADE: moderate) (Figure 5). As the regressive analysis for death and composite cardiovascular events (Appendix A) showed, the baseline LDL-c value did not exert a significant influence on the comparisons among those medications. With no trial informing the direct comparisons between PS and C within the network, we failed to perform the regression analysis for the corresponding result. After excluding the trials with established ASCVD patients occupied >70%, the limited number of studies hamper the conduct of NMA while the pairwise meta-analysis provided the consistent effect of statin-based treatment (Appendix A). Using the frequentist model, including or excluding the trials with a high risk of bias, the results remained consistent with those in the analysis with the Bayesian model (Appendix A).

## 4. Discussion

This systematic review and NMA summarized the latest evidence for the benefits and safety of current antidyslipidaemic pharmacotherapies among moderate-advanced CKD. Compared with the previous pairwise meta-analysis, NMA compared the effects of various medicines and provided more information. In addition to the benefits of statins (with or without ezetimibe) [4,6,7,56,57], this NMA further concluded the value of other hypolipidemic agents, especially PCSK9i. All 18 RCTs involving 41,213 participants provided moderate certainty evidence that compared to placebo or no hypolipidaemic therapy, PS, HS, and LS were associated with a significant reduction in all-cause mortality among CKD patients. For long-term composite cardiovascular events, moderate certainty evidence provided by 20 trials enrolling 42,286 CKD patients supported the benefit of PS, HS, LS, and F on risk reduction. Low certainty evidence supported the significant association between ES and composite cardiovascular events reduction. These pharmacotherapies benefit CKD patients without leading to serious adverse events or a negative influence on renal function. Of note, studies of low-risk indicated that statins might exert a dose-dependent beneficial effect on renal function. Regarding the application of other traditional agents, we concluded that colestilan was an effective and well-tolerated treatment for LDL-c reduction in CKD even in the dialysis group, without evidence supporting long-term cardiovascular risk reduction. Apabetalone was expected to dramatically improve the prognosis of CKD patients.

Statins effectively reduced all-cause death and composite cardiovascular events in CKD, which was consistent with the findings of previous meta-analyses and the recommendations for statin application in CKD [6]. In addition, our work provided some other important information. Firstly, compared to previous works enrolling all CKD patients including those with eGFR of 60–90 mL/min/1.73 m^2^, our study supported the consistent treatment effect in moderate-advanced stage patients. Secondly, in contrast to the opinion in the KDIGO Clinical Practice Guideline that HS in CKD was not recommended given its uncertain safety [6], our study indicated that this prescription might be associated with a greater magnitude of risk reduction, with no or few serious adverse events and a negative influence on renal function. Being characterized as having a high/very high risk of ASCVD, CKD patients may reap additional benefits from aggressive statin therapy. However, the value of statins in primary prevention requires further serious assessment since those receiving aggressive treatments enrolled in our work mostly had prior ASCVD. Thirdly, the benefits of statins may not be limited to cardiovascular protection, as their dose-dependent renoprotective properties have been reported by an increasing number of investigators [18,19,58,59]. This is encouraging because a downward trend should otherwise have been expected based on the natural history of CKD [60]. The factors accounting for this phenomenon are not well known, in which vascular protection, lipids, CRP reduction, and a decrease in proteinuria may play important roles. However, the absence of Asian participants in those trials influences the generalization to a broader population owing to differences in lipid profiles and medication tolerance. While the effects of statins in protecting cardiovascular health were blunted in dialysis according to the subgroup analysis, indicating that the use of these medications should not be extended to dialysis patients given the uncertain benefit and cumulative drug toxicity. Their cardiovascular events are proportionally increased due to non-atheromatous processes such as sudden death and heart failure [4,7], which would not be directly improved by hypolipidemic treatments [61,62]. However, the limited number of patients enrolled might be underpowered to detect the difference. Moreover, the conventional low-dose statin therapy given the safety among them may not be the optimal choice to maximize benefits. Therefore, the value of lipid-lowering intervention in dialysis patients requires more investigation, especially for patients with additional cardiovascular risks such as existing CAD and higher LDL-c [63].

PS had the highest probability of reducing the occurrence of all-cause death and composite cardiovascular events, suggesting that the addition of PCSK9i was probably associated with further risk reduction. This action may be explained by the dramatic effect on lipid management, especially the potency of reducing LDL-c by approximately 60% on top of statin (±ezetimibe) use. For CKD patients, the cardiovascular risk increases as the eGFR declines, and thus, additional timely intervention is required to enhance cardiovascular protection. However, treatment escalation with a double-dose statin can produce only an incremental LDL-c reduction of 6%, and the clinical benefits are attuned to renal disease progression [64]. Moreover, serum PCSK9, which is inversely correlated with eGFR, was roughly double in CKD patients compared with healthy controls [65]. In this context, PCSK9i is likely to provide additional cardiovascular benefits and has a unique advantage compared to statin monotherapy. However, the extensive use of PCSK9i in practice cannot be achieved currently due to their perceived expense and the limited amount of evidence. Based on the ‘highest risk–highest benefit’ strategy and related published trials, this powerful antidyslipidaemic agent is considered cost-effective only in patients with clinical atherosclerotic cardiovascular disease and other comorbidities [4,66]. More clinical information assessing the efficacy and safety profile is needed before PCSK9-targeted therapy can be recommended to CKD patients for primary prevention. Furthermore, a critical question is whether PCSK9i can remain effective in preventing adverse cardiovascular events as kidney impairment progresses, even in dialysis. Our analysis further stresses the additional value of PCSK9i in CKD and indicates the need for investigators to assess the cost-effectiveness of wide application in the general CKD population.

The combination of statin and ezetimibe failed to lower death risk effectively, which might be because the CKD patients enrolled in SHARP, the key source of evidence related, were mostly at stage4–5 and even dialysis individuals with a very high risk of death induced by non-atherosclerosis-related diseases. Currently, the analysis of ES in CKD was restricted to a limited number of related RCTs. Although it was proven to be associated with the reduction of composite cardiovascular events, the grade of evidence was low. Therefore, the investigation of ES, especially the comparisons between ES and statin monotherapy in CKD, is needed urgently. And the optimality of this prescription in guidelines should be re-evaluated with the progress of antidyslipidemia pharmacotherapy.

Fibrates with/without statins could effectively further improve lipid parameters (especially TG and HDL-c) and reduce total cardiovascular events [15,33,47,67] among CKD persons. CKD persons are frequently characterized as having high TG but normal LDL-c levels, which might explain their benefit, as their effect on cardiovascular-related outcomes among general persons had been proven to be equivocal and more evident in those with higher TGs/lower HDL-c [4,68]. However, the limited number of studies included in our work made this evidence less robust and the studies on dialysis patients were absent. Therefore, the CV benefits require more confirmation and the ongoing PROMINENT trial may provide additional information [69]. As for the application of the other traditional agents and fish oil, the sparsity of evidence hampered the investigation of the association between them and risk reduction. Apabetalone was expected to dramatically improve the prognosis of CKD patients [55].

Hypolipidemic medications protect cardiovascular health through pleiotropic pharmacological mechanisms, such as lipid reduction effects and the stabilization of atherosclerotic plaques. However, there is conflicting information on the explanations accounting for their cardiovascular protective effects in CKD regarding their unique pathophysiological characteristics. First, the link between serum lipid levels and the cardiovascular outcome is not clear in CKD, with debates surrounding the role of LDL-c, which is often within normal limits [3]. The relation between LDL-c and cardiovascular risk was deemed less robust and the KDIGO guidelines rejected treat-to-target goals [6]. However, there were still several large trials reporting the association between per 1 mmol/L LDL-c and overall risk reduction [23], and the magnitude of risk reduction among these medications remained roughly consistent with that of LDL-c reduction in our work. Except for the discussion on LDL-c, the effect of other components of lipid profiles, especially TG and HDL-c with reversed functions, has also drawn attention in recent years [70]. Second, inflammation is a key process observed in CKD and has been shown to predict the long-term risk of developing atherosclerotic vascular disease events and death. Lipid-lowering agents, particularly statin-based therapy and PCSK9i, could protect cardiovascular health by attenuating the inflammatory response [19,31,32,71,72,73,74,75] and achieve the largest effect on relative risk reduction among patients who reduced not only LDL-c by more than 50% but also hs-CRP levels by more than 50% [19]. Last, the effects of reducing proteinuria, a potential risk factor for CAD, may contribute to the decrease in cardiovascular events, and this benefit was independent of the influence on lipids [76].

This work had some limitations. (1) Although this study was based on the relevant articles of the highest quality of evidence available currently, the results should be treated seriously since some data obtained from reported post hoc analyses of RCTs might be less reliable. We will update this work accordingly when additional valuable data become available. (2) There was an imbalanced distribution of participants among treatment strategies. As the network plots showed, most included trials studied the effects and safety of low-intensity statins, while the trials focusing on other drug regimens were limited.

## 5. Conclusions

In conclusion, this systematic review and Bayesian NMA provide a comprehensive overview of the state of the evidence available concerning the effects and safety of hypolipidemic medications in CKD patients. Despite comorbid ASCVD and LDL-c levels, statin-based treatment could effectively reduce long-term all-cause mortality and composite cardiovascular events in nondialysis CKD populations. Aggressive therapies, such as PS and HS, appeared to be more effective, without causing major adverse events or further impairment of renal function. In addition to their cardioprotective value, statins were reported to have potential dose-dependent renoprotective properties. The present work was an update of the previous meta-analysis, and the first one to summarize and rank the effect of various lipid-lowering drugs in CKD, with the expectation to provide some reference for clinical practical and further research.

## Figures and Tables

**Figure 1 pharmaceutics-15-00006-f001:**
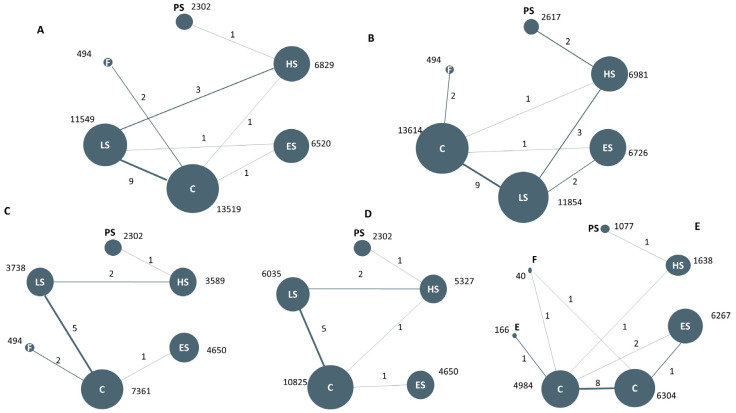
Network comparison of different antidyslipidemia pharmacotherapies for the clinical outcomes: All-cause death (**A**), composite cardiovascular events (**B**), cardiovascular death (**C**), stroke (**D**), LDL-c reduction (**E**). Note: The circle size represents the sample size of the group; the thickness of the edge refers to the number of studies. PS: PCSK9 inhibitors plus statin; HS: moderate-high intensity statins; LS: low-moderate intensity statins; ES: ezetimibe plus statins; F: fibrates; C: placebo or no lipid-lowering treatment.

**Figure 2 pharmaceutics-15-00006-f002:**
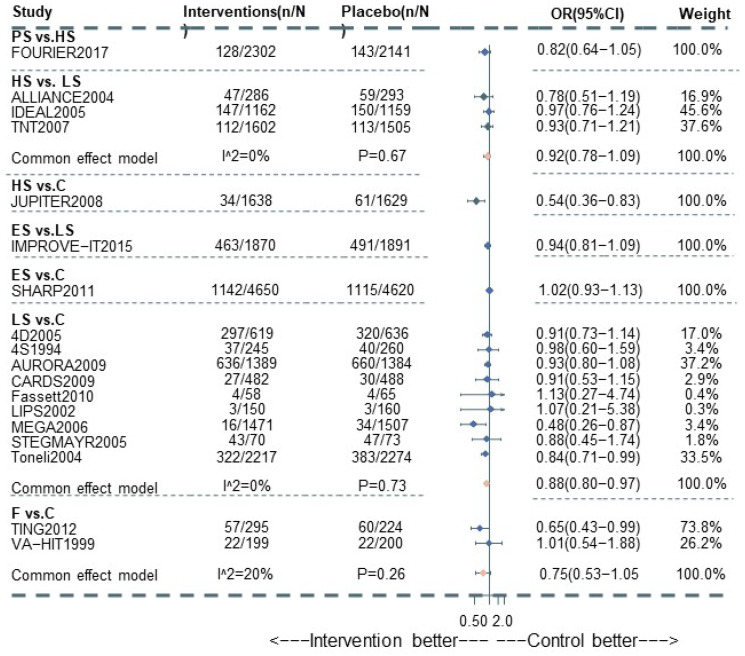
Pairwise meta-analysis for All-cause death among CKD patients.

**Figure 3 pharmaceutics-15-00006-f003:**
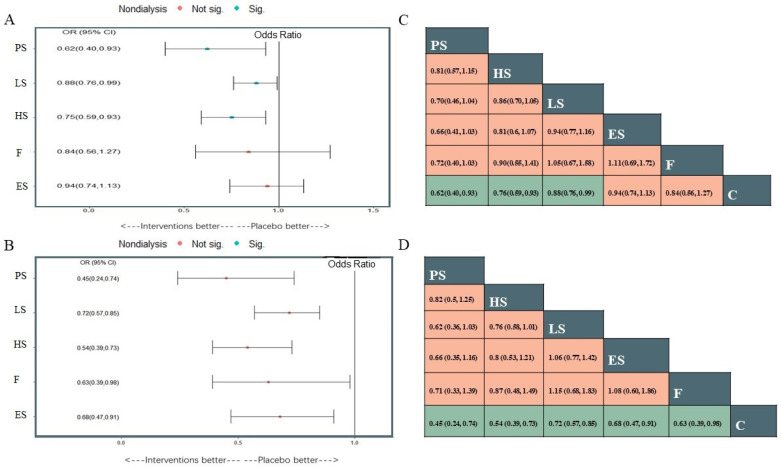
Plots of different treatments compared with placebo or no lipid-lowering medication use for All-cause death (**A**) and the composite cardiovascular events (**B**), and the league tables of the two outcomes analysis for the network estimates of all comparisons (**C**) all-cause death; (**D**) the composite cardiovascular events). Sig: the difference was statistically significant; Not sig: the difference was not statistically significant.

**Figure 4 pharmaceutics-15-00006-f004:**
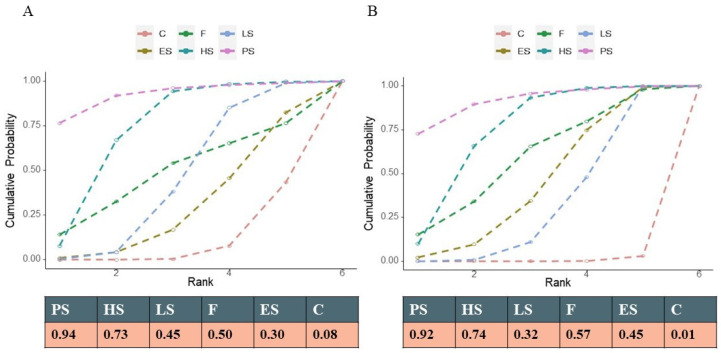
Cumulative rank probability. Note: The area under the curve represents the cumulative rank probability of each treatment. (**A**) All-cause death; (**B**) the composite cardiovascular events.

**Figure 5 pharmaceutics-15-00006-f005:**
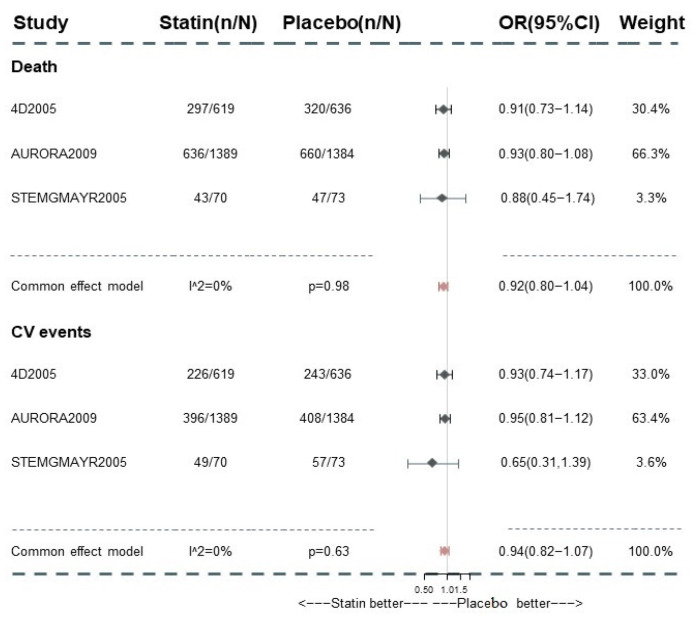
Subgroup analysis for all-cause death and composite cardiovascular event in dialysis patients. CV events: composite cardiovascular events.

**Table 1 pharmaceutics-15-00006-t001:** Baseline characteristics of studies included in the NMA.

Study	Sample Size	Follow-Up (Month)	CKD Definition	I	C	Age (Years)	Male (%)	Smoker (%)	ASCVD (%)	Diabetes (%)	Re. (%)	Dialysis (%)	LDL-c (mmol/L)	HDL-C (mmol/L)	TG (mmol/L)	SCr (umol/L)
Charytan 2019 [16] (FOURIER trial)	4443	30.0	eGFR20–59 mL/min/ 1.73 m^2^	PS	HS	68.7 (7.8)	65.0	15.8	100: MI 77.0; Stroke 25.0; PAS 17.4	46.4	NA	0	2.4 [2.1–2.7]	1.1 [0.9–1.4]	1.6 [1.2–2.2] *	114.9 [106.1–132.6]
Toth 2018 [17]	467	6–26.0	eGFR 30–59 mL/min/ 1.73 m^2^	PS	HS	67.3 (9.4)	57.0	57.9	NA	48.2	NA	0.0	2.8 (0.9)	1.3 (0.3)	1.5 (0.8)	115.5 (NA)
Hagiwara 2017 (HIJ-PROPER trial) [14]	424	46.0	eGFR < 60 mL/min/1.73 m^2^	ES	LS	65.6 (11.8)	75.6	34.5	100.0	30.2	NA	NA	3.5 (0.8)	NA	NA	NA
Baigent 2011 (UK-SHARP-I trial) [23]	9270	59.0	More than once previous measurement of serum or plasma creatinine of ≥150 μmol/L in men or 130 μmol/L in women.	ES	C	62.0 (12.0)	62.5	13.0	No history of MI or Re. Prior vascular disease: 15.0	23.0	0	32.6	2.8 (0.9)	1.1 (0.3)	2.3 (1.7)	At least 150.0
Stanifer 2017 (IMPROVE-IT trial) [22]	3761	84.0	eGFR < 60 mL/min/1.73 m^2^	ES	LS	70.8 (9.3)	39.5	19.9	MI 27.6	33.8	PCI 12.1; CABG 15.1	0	2.3 (0.5)	1.1 (0.4)	1.5 (0.8)	123.8 (26.5)
Tonelli 2004 (VA-HIT trial) [35]	399	61	eGFR < 67.5 mL/min/1.73 m^2^	F	C	67.3 (5.2)	100.0	NA	100.0	30.3	NA	NA	2.9 (0.6)	0.8 (0.1)	1.9 (0.8) *	Less than 176.0
TING 2012 [34]	519	60	eGFR 30–59 mL/min/ 1.73 m^2^	F	C	66.5 (5.9)	41.2	6.9	39.7	100.0	NA	0.0	3.1 (NA)	1.1 [1.0–1.1]	1.9 [1.9–2.0] *	NA
Shepherd 2007 (TNT trial) [21]	3107	60	eGFR < 60 mL/min/1.73 m^2^	HS	LS	65.5 (7.0)	67.7	9.0	100.0	17.5	>50.0	NA	2.5 (0.5)	1.2 (0.3)	1.8 (0.8)	NA
Koren 2008 (ALLIANCE trial) [20]	579	54.3	eGFR < 60 mL/min/1.73 m^2^	HS	LS	65.2 (7.2)	76.7	15.0	100.0	27.9	>50.0	NA	3.8 (0.7)	1.0 (0.3)	2.3 (1.1)	136.2 (44.2)
Holme 2009 (IDEAL trial) [18]	2321	58.0	eGFR < 60 mL/min/1.73 m^2^	HS	LS	67.0 (7.9)	NA	15.3	100.0	14.4	38.3	NA	3.2 (1.0)	1.2 (0.3)	1.7 (0.9)	NA
Ridker 2010 (JUPITER trial) [19]	3267	22.8	eGFR < 60 mL/min/1.73 m^2^	HS	C	70.0 [65.0–75.0]	34.8	8.1	0.0	NA	0.0	NA	2.8 [2.5–3.1]	1.3 [1.0–1.6]	1.5 [1.1–2.1] *	NA
Wanner 2005 (4D trial) [34]	1255	48.0	Undergoing hemodialysis	LS	C	65.7 (8.3)	54.0	8.6	CAD:29.4; PAD 44.6	100	13.0	100	3.3 (0.8)	0.9 (0.4)	3.1 (1.9)	NA
Tonelli 2004 (3p: WOSCOPS, CARE, LIPID) [44]	4491	60.0	CG-GFR 30–59 mL/min/ 1.73 m^2^	LS	C	65.7 (5.6)	81.7	10.3	Previous stroke5.3; known CAD73.7; MI67.6; UA21.2	9.9	NA	0.0	3.9 (0.7)	1.0 (0.2)	1.8 (0.8)	115.0 (17.7)
Lemos 2005 (LIPS trial) [36]	310	45.6	Mild Renal Impairment: abnormal creatinine clearance was defined as a value in the lowest quintile (<55.9 mL/min)	LS	C	69.0 (7.0)	67.0	17.0	100: patients who underwent successful elective PCI were included	12.0	100.0	0.0	3.4 (0.8)	1.0 (0.3)	1.7 (0.8) *	117.6 (247.5)
STEGMAYR/ HOLMBERG 2005 [37]	143	31.0	Creatinine clearance < 30 mL/min	LS	C	68.6 (11.3)	69.3	61.0	Angina25.9, MI24.5, stroke or TIA 12.6	26.8	7.7	76.9	3.5 (1.2)	1.2 (0.5)	2.6 (1.7)	NA
Chonchol 2006 (4S trial) [38]	505	65.5	CG-GFR 30–59 mL/min/ 1.73 m^2^	LS	C	60.4 (6.2)	73.3	20.3	100.0	4.1	7.7	NA	4.9 (0.7)	1.2 (0.3)	1.5 (0.5) *	NA
Kendrick 2009 (AFCAPS/ TexCAPS trial) [39]	304	63.6	eGFR < 60 mL/min/1.73 m^2^	LS	C	62.0 (7.5)	78.3	7.9	0.0	1.5	0.0	0.0	3.9 (0.6)	1.0 (0.2)	2.0 (0.9) *	123.8 (26.5)
Nakamura 2009 (MEGA trial) [40]	2978	63.6	30 < eGFR < 60 mL/min/1.73 m^2^	LS	C	58.0 (7.2)	24.3	12.5	0.0	NA	0.0	0.0	4.0 (0.4)	1.5 (0.4)	1.5 [1.1–2.1]	NA
Colhoun 2009 (CARDS trial) [41]	970	46.8	30 < eGFR < 60 mL/min/1.73 m^2^	LS	C	65.0 (6.7)	47.9	NA: as on the criteria of inclusion	0.0	100.0	0.0	0.0	3.1 (0.7)	1.5 (0.2)	NA	113.2 (NA)
Fassett 2009 [42]	123	30.0	serum creatinine levels >120 mmol/L	LS	C	60.2 (15.1)	65.2	NA	NA	8.5	NA	0.0	3.4 (1.1)	1.2 (0.4)	2.3 (1.5) *	226.1 (118.2)
Tuñón 2020 (ODYSSEY OUTCOMES trial) [45]	2122	36.0	eGFR20–59 mL/min/ 1.73 m^2^ (CKD-EPI)	PS	HS	67.1 (8.9)	59.2	11.8	MI 26.2; stroke 16.3; PAD 8.5	41.4	34.5	0.0	2.3 [1.9–2.8]	1.1 [0.9–1.3]	1.6 [1.2–2.2]	NA
Suzuki 2014 [24]	286	12.0	eGFR < 60 mL/min/1.73 m^2^ (MDRD)	ES	LS	64.0 (12.0)	66.0	40.2	Cerebrovascular 6.9; cardiovascular 2.8; PAD 4.2	34.5	NA	0.0	3.3 (0.2)	1.5 (0.2)	NA	NA
Kimura 2017 (ASUCA trial) [25]	334	24.0	1)With positive proteinuria and eGFR >60 (ml/min/1.73 m^2^); 2)30 < eGFR <60 mL/min/1.73 m^2^ (MDRD)	LS	Non-statin treatment	63.2 (8.1)	63.8	14.1	Cerebrovascular accident 5.7; MI:0.6; angina pectoris1.2	33.8	NA	0.0	3.7 (0.7)	1.3 (0.3)	2.0 (1.4)	NA
Dogra 2007 [27]	90	1.5	eGFR < 60 mL/min/1.73 m2 (Cockcroft-Gault formula)	Tow arms: F and LS	Placebo	61.9 (12.9)	62.2	0.2	0.2	21.1	NA	41.1	3.3 (1.1)	1.2 (NA)	1.8 (NA)	NA
Nanayakkara 2007 (ATIC trial) [28]	93	Maximum 24.0	Creatinine clearance of 15 to 70 mL/min per 1.73 m^2^ (MDRD)	LS	C	53.0 (12.0)	56.9	35.5	0.0	0.0	0.0	NA	3.6 (0.9)	1.2 (0.4)	1.8 (1.0) *	205.1 (83.9)
Yasuda 2004 [30]	80	Maximum 12.0	Scr < 440 mmol/L or creatinine clearance was 20–70 mL/min/1.73 m^2^ (urinary creatinine concentration × urine volume)/(SCr concentration).	LS + diet	Diet	57.5 (2.0)	46.3	NA	NA	42.5	NA	0.0	4.4 (0.1)	1.3 (0.1)	2.5 (0.2) *	168.0 (17.7)
Goicoechea 2006 [31]	63	6.0	CKD stages 2, 3, and 4: eGFR< 90 and >15 mL/min./1.73 m^2^ (Cockroft-Gault formula.)	LS	C	67.3 (13.8)	63.5	NA	PAD: 11.1; Cerebrovascular disease: 3.2; CAD 3.2	17.5	0.0	0.0	3.7 (0.6)	1.5 (0.4)	1.6 (0.7) *	NA
Verma 2005 [32]	91	5.0	eGFR < 60 mL/min/1.73 m^2^ (MDRD)	LS	C	73.5 (14.9)	35.2	19.0	NA	46.5	NA	0.0	3.4 (1.0)	1.3 (0.4)	1.9 (1.6) *	141.4 (88.4)
Burmeister 2009 [43]	59	3.0	Hemodialysis	LS	C	57.1 (15.4)	62.7	NA	Acute MI was excluded	33.9	NA	100.0	2.4 (1.0)	1.0 (0.3)	1.9 (0.9) *	NA
Fellström 2009 (AURORA trial) [29]	2773	45.6	undergoing hemodialysis	LS	C	64.2 (8.6)	62.0	15.5	CVD39.9, Peripheral vascular disease 15.3	19.3	6.3	100.0	2.6 (0.9)	1.2 (0.4)	1.8 (1.1)	NA
**Summary**	**45,627**	**51.8**	**/**	**/**	**/**	**65.4 (10.7)**	**58.0**	**14.0**	**/**	**30.9**	**/**	**20.8**	**3.0**	**1.2**	**1.9**	**/**

NOTE: When reported, outcomes were extracted based on the intention-to-treat (ITT) principle. Whenever possible, adjusted outcomes were extracted. When available, additional information was acquired from study protocols and appendices. CKD: chronic kidney disease; eGFR: estimated glomerular filtration rate; CKD-EPI: Chronic Kidney Disease Epidemiology Collaboration equation; MDRD: Modification of Diet in Renal Disease equation; ASCVD: atherosclerotic cardiovascular disease; I: intervention; C: control group; Re. revascularization; LDL-c: low-density lipoprotein cholesterol; HDL-c: high-density lipoprotein cholesterol; SCr: serum creatinine; MI: myocardial infarction; PAD: Peripheral arterial disease; PS: proprotein convertase subtilisin/Kexin type 9 inhibitor and the background use of statins; HS: high-intensity statins; LS: low-intensity statins; ES: Ezetimibe plus statin; F: fibrates; C: placebo/diet therapy alone/ blank control; ACEI: angiotensin receptor antagonist; ARB: Angiotensin II receptor antagonist; NA: not available; SD: standard deviation.; *: fasting plasma triglyceride level. Data are expressed as n (%), mean (SD), or median [quartile 1–quartile 3].

## Data Availability

The analytic dataset is available on request by contacting the corresponding author.

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
