# Peer review of "Antidyslipidemia Pharmacotherapy in Chronic Kidney Disease: A Systematic Review and Bayesian Network Meta-Analysis"

_pharmaceutics, 2022, doi:10.3390/pharmaceutics15010006_

Round 1

Reviewer 1 Report

First, I would like to express my gratitude to the editor who invited me to be a reviewer in an excellent journal, pharmaceutics.

As I am mainly researching evidence synthesis methodology-based data mining, I was able to review the manuscript for review with great interest.

After reviewing the manuscript over the given period, I have come to the following opinion.

Major opinion

- At least based on the methodological frame of NMA, I did not find any serious methodological flaws in this manuscript. All analyzes were performed according to reliable procedures. There may be disagreements about the analysis results, but that will not work as a reason why this manuscript will not be published.

minor suggestion

- NMA, not a pairwise meta-analysis, requires a suitable confidence evaluation tool. As a tool optimized for NMA has been proposed very recently, it is recommended to further improve the reproducibility of the manuscript by utilizing it. Use the references below in your work.

[Suggested ref.] Nikolakopoulou A, Higgins JPT, Papakonstantinou T, Chaimani A, Del Giovane C, Egger M, Salanti G. CINeMA: An approach for assessing confidence in the results of a network meta-analysis. PLoS Med. 2020 Apr 3;17(4):e1003082. doi: 10.1371/journal.pmed.1003082.

It's a short opinion, but I hope it will be helpful to the authors. I look forward to rereading it in detail once the publication is complete.

Author Response

Dear Reviewer:

On behalf of my co-authors, I am very grateful to you for giving suggestions on our manuscript. CINeMA is a useful approach for assessing confidence in the results of a network meta-analysis. Actually, we noticed this tool and applied it to assess confidence in our results before. However, CINeMA calculates the contribution of studies in NMA and the treatment effects by using R packages meta and netmeta, which apply the frequentist approach but not the Bayesian model. To our knowledge, whether CINeMA can be applied to the analysis of Bayesian NMA is still controversial. Therefore, we finally removed this part and completed our study utilizing the GRADE method. Considering your valuable suggestion seriously, we authors discussed this issue and consulted relevant experts, but we failed to reach a consensus. In this context, we would like to leave the decision to the editor and reviewer. We mention this assessment approach in the Section Method (revised manuscript page 5, line 121) and provide the assessment results utilizing CINeMA as a separate supplementary file (supplementary file 2). If the editor and reviewer finally consider this part unfit for publication, we are willing to remove it.

Please do not hesitate to contact us if there is any question. Thanks again for your hard work. 

Reviewer 2 Report

The paper reports a systematic review with meta-analysis of clinical controlled trials (RCTs). The authors performed a Bayesian Network Meta-analysis (BMA). The issue addressed by the authors has clearly  hot nature- the role of antidyslipidemic agents in the progression of CKD and death risk (including dialysis population). I have some minor comments:

1) the authors should add in the Section Discussion a brief comment regarding the advantages of BMA in comparison with conventional meta-analysis of RCTs (only a few sentences)

2) Line 53 'CKD has been a public health problem' this is incorrect. I suggest to write 'CKD is currently a public health problem'

3) lines 140-143 there are two sentences and I suggest to rewrite these as these are confusing

3)  Lines 352-4 The sentence regarding Limitation 1 is unclear and I suggest to rewrite it

4) Line 207 'not enough data were available for MI analysis' please let me klnow the meaning of MI 

5) The quality of Figure 1 is not satisfactory and I suggest to delete (or change) it 

Author Response

Dear Reviewer:

Thanks for your constructive feedback and we have fully revised our manuscript according to the suggestions. The major revisions and the responses to your comments are as follows:

Response 1. In the revised manuscript, page 9, line 260, we have added a brief comment regarding the advantages of BMA in comparison with conventional meta-analysis of RCTs.

Response 2. In the revised manuscript, page 3, line 53, we have rewritten the sentence according to your suggestion.

Response 3. In the revised manuscript, page 5, line 140, we have rewritten the sentence to make our idea clearer.

Response 4. MI is the abbreviation for myocardial infarction. We apologize for our negligence and we added the words before the abbreviation in the revised manuscript, page 7, line 209.

Response 5. Thanks for your suggestion and we have modified Figure 1 to make this result clearer.

Special appreciation to your good comments.